# Delay-DSGN: A Dynamic Spiking Graph Neural Network with Delay Mechanisms for Evolving Graph

**Zhiqiang Wang** [1]  **Jianghao Wen** [1]  **Jianqing Liang** [1]

## Abstract

Dynamic graph representation learning using Spiking Neural Networks (SNNs) exploits the temporal spiking behavior of neurons, offering advantages in capturing the temporal evolution and sparsity of dynamic graphs. However, existing SNN-based methods often fail to effectively capture the impact of latency in information propagation on node representations. To address this, we propose Delay-DSGN, a dynamic spiking graph neural network incorporating a learnable delay mechanism. By leveraging synaptic plasticity, the model dynamically adjusts connection weights and propagation speeds, enhancing temporal correlations and enabling historical data to influence future representations. Specifically, we introduce a Gaussian delay kernel into the neighborhood aggregation process at each time step, adaptively delaying historical information to future time steps and mitigating information forgetting. Experiments on three large-scale dynamic graph datasets demonstrate that Delay-DSGN outperforms eight state-of-the-art methods, achieving the best results in node classification tasks. We also theoretically derive the constraint conditions between the Gaussian kernel's standard deviation and size, ensuring stable training and preventing gradient explosion and vanishing issues.

## 1. Introduction

Graph representation learning aims to map graph data, including nodes, edges, or entire graphs, into a low-dimensional vector spaces for downstream tasks such as node classification and link prediction. Real-world graphs, however, are often dynamic, requiring dynamic graph representation learning methods to account for the temporal dependencies of nodes and edges in order to capture the graph's evolving nature and generate more accurate representations. This ability is particularly important in dynamic applications such as social network analysis and traffic flow prediction.

In recent years, researchers have proposed various methods to address the dynamic nature of graphs. Random walk-based methods such as CAWs (Wang et al., 2021) and NeurTWs (Jin et al., 2022) capture contextual information from dynamic graphs by sampling random walk sequences of nodes, thereby enabling node encoding. Methods using graph neural networks (GNNs) aggregate information from neighboring nodes based on the graph topology, often incorporating different temporal modeling strategies to capture dynamic graph evolution. For instance, approaches like JODIE (Kumar et al., 2019), GAEN (Shi et al., 2021) and CHNN (Yin et al., 2024a) use recurrent neural networks (RNNs) (Cho et al., 2014) to model temporal dependencies in graphs, while HTNE (Zuo et al., 2018), M$^2$DNE (Lu et al., 2019), and TREND (Wen & Fang, 2022) treat time as a continuous variable, modeling event timings through temporal point processes. However, these methods often require extensive sequence extraction or large-scale memory units, leading to low modeling efficiency in large-scale dynamic graphs.

As a class of low-power neural networks (Maass, 1997; Schliebs & Kasabov, 2013; Pfeiffer & Pfeil, 2018), Spiking Neural Networks (SNNs) utilize Poisson rate coding to convert continuous node features into sparse binary representations, offering significant advantages in modeling efficiency for large-scale graphs. Specifically, SNN neurons simplify internal computation through linear integration, avoiding complex nonlinear activations and matrix multiplications. Additionally, SNNs can capture temporal evolution patterns by accumulating and memorizing historical states, extending beyond spatial neighborhood-based information transfer. Recent works, such as SpikeGCN, STFN, SpikeNet, and Dy-SIGN (Zhu et al., 2022; Xu et al., 2021; Li et al., 2023; Yin et al., 2024b), combine SNNs with GNNs, employing

---

[1] Key Laboratory of Computational Intelligence and Chinese Information Processing of Ministry of Education, School of Computer and Information Technology, Shanxi University, Taiyuan 030006, Shanxi, China. Correspondence to: Jianqing Liang <liangjq@sxu.edu.cn>.

*Proceedings of the 42$^{nd}$ International Conference on Machine Learning*, Vancouver, Canada. PMLR 267, 2025. Copyright 2025 by the author(s).

graph convolution layers with SNN neurons to aggregate neighborhood information and alleviate the computational complexity associated with matrix multiplications and non-linear activations in GNNs.

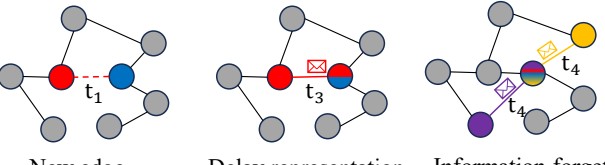

| New edge | Delay representation | Information forget |

*Figure 1.* Illustration of information dynamics in a social network. At time $t_1$, two nodes establish a connection without any inter-action, thus having no immediate impact on their representations (depicted by unchanged colors). At time $t_3$, the two nodes begin exchanging information, leading to changes in their representations (depicted by color updates). Finally, at time $t_4$, additional nodes participate in information exchange, where node representations are influenced by both current interactions and decayed historical information.

Despite progress in dynamic graph representation learning, significant challenges remain in accurately modeling the influence of historical information on current and future node representations, particularly with respect to delay representation and historical information forgetting. Existing methods typically update node representations by aggregating current-time information, ignoring the possibility that such information may impact node representations at later time steps. This phenomenon, known as delay representation, can be illustrated by the following example in social networks in Figure 1: at time point $t_1$, a new edge is added between nodes A and B, but the edge does not immediately affect the node representations of A and B; instead, the impact manifests at a later time step. This delay effect reflects the inherent latency in real-world information propagation. Moreover, as dynamic graphs evolve, node representations are progressively updated, and earlier historical information tends to be overshadowed or forgotten due to the influence of new information. For example, over time, the representation of node A may be influenced by several neighboring nodes, causing early neighbors' information to be diluted or lost, thereby erasing valuable historical context. Therefore, effectively modeling delay representation and mitigating historical information forgetting are key challenges in dynamic graph representation learning.

Inspired by the synaptic plasticity of biological neurons (Izhikevich, 2006; Bowers, 2017; Hammouamri et al., 2024), we address the problems of delay representation and historical information forgetting in dynamic graph representation learning by introducing a synaptic delay mechanism. Synaptic delay refers to the time it takes for information to propagate between neurons, which varies due to synaptic

heterogeneity. With this mechanism, changes in topology do not immediately affect node representations but take effect after a certain delay. Meanwhile, the presence of this delay allows historical information to influence node representations at the current time step, effectively alleviating the problem of information forgetting. Neurons can leverage this heterogeneous delay mechanism to recognize complex spatiotemporal spike patterns, thereby effectively capturing the influence of historical information on current and future node representations.

To this end, we propose Delay-DSGN, a dynamic spiking graph neural network with a learnable delay mechanism. Delay-DSGN integrates the synaptic delay mechanism into SNNs, so that the aggregated node features are not instantaneously transmitted to the neurons. Instead, the model adaptively learns appropriate delay times between neurons through a Gaussian delay kernel, enabling delay representation of features. This effectively mitigating the forgetting of historical information and generating more accurate node representation. Our main contributions are as follows:

- We propose Delay-DSGN, a dynamic spiking graph neural network with a learnable delay mechanism, which dynamically adjusts connection weights and spike propagation speeds to achieve more accurate modeling of dynamic graph evolution.

- We theoretically derive the constraint conditions between the standard deviation and size of the Gaussian delay kernel, ensuring that the Delay-DSGN model avoids gradient explosion and vanishing problems during training.

- Extensive experiments validate that Delay-DSGN outperforms current state-of-the-art methods in node classification tasks across multiple dynamic graph datasets.

## 2. Related Work

### 2.1. Shallow Representation

Early dynamic graph representation learning focused on shallow encoding methods like matrix/tensor decomposition and random walks, without leveraging the deep encoding capabilities of graph topology. Matrix decomposition methods, such as incremental SVD (Chen & Tong, 2015) and error-bound restart (Zhang et al., 2018), perform joint factorization of adjacency matrices over multiple time snapshots to obtain low-rank representations. Tensor decomposition introduces a time dimension into the adjacency matrix, forming a tensor to capture temporal evolution. Common approaches include CP and Tucker decomposition (Acar & Yener, 2009). In contrast, random walk methods generate sequences of random walks from nodes (i.e., contextual information) to learn representations, capturing higher-order de-

pendencies between nodes. For example, (De Winter et al., 2018) and (Sandra & Jochen, 2019) employed node2vec (Grover & Leskovec, 2016) and a variant of DeepWalk (Perozzi et al., 2014) for random walks over each network snapshot. Other related methods include DNE (Chen et al., 2019), tNodeEmbed (Singer et al., 2019), and DynSEM (Yu et al., 2017). CTDNE (Nguyen et al., 2018), incorporates temporal factors directly into the sampling process, learning embeddings with temporal walk sequences.

## 2.2. GNN-Based Representation

GNNs leverage graph topology to encode node representations, enhancing the learning of structural information. By integrating RNNs, static GNNs are extended for dynamic process modeling. GC-LSTM (Chen et al., 2022) and Dyngraph2Seq (Gao et al., 2019) combine GCN (Kipf & Welling, 2017) with LSTM (Hochreiter & Schmidhuber, 1997). TGN (Rossi et al., 2020) and JODIE (Kumar et al., 2019) update node hidden states using RNNs units for representation learning. EvolveGCN (Pareja et al., 2020) adjusts GNNs weights over time using RNNs to capture dynamic information in evolving networks. Additionally, TGAT (Xu et al., 2020) introduces temporal encoding and self-attention mechanisms to aggregate neighbor node features across time, enhancing the understanding of topological changes in dynamic graphs. DyRep (Trivedi et al., 2019), based on temporal point processes, captures both graph-level and node/edge-level temporal variations. TREND (Wen & Fang, 2022) uses Hawkes processes to model the impact of historical events on current ones, analyzing neighborhood formation effects on nodes through Hawkes conditional intensities.

## 2.3. SNN-based Representation

To address the high computational cost of dynamic graph representation learning, researchers have explored applying low-power SNNs to graph data. STFN (Xu et al., 2021) and SpikingGCN (Zhu et al., 2022) combine GCN with SNNs, where GCN aggregates neighborhood information, and SNNs encode node features as spike signals, simulating time steps. SpikeGCL (Li et al., 2024) links graph contrastive learning with spike. These methods achieve promising results on static graphs. Additionally, SpikE (Dold & Garrido, 2021), R-GCN (Chian et al., 2021), and GRSNN (Xiao et al., 2024) use spike for knowledge graphs, representing relationships as spike time differences. Recent work has extended SNNs to dynamic graphs, such as SpikeNet (Li et al., 2023) and Dy-SIGN (Yin et al., 2024b). These approaches leverage the dynamic characteristics of SNNs, where each graph snapshot corresponds to a time step in the SNN, capturing the temporal evolution of graphs. Dy-SIGN extends the concept of implicit fixed-point equilibrium to graph learning (Gu et al., 2020), addressing evaluation and

training issues in recurrent GNNs, further reducing memory consumption.

## 3. Problem Definition

Given a sequence of dynamic graph snapshots $\{G^1, G^2, \ldots, G^T\}$, where $T$ represents the total number of time steps in the dynamic graph, each snapshot is defined as $G^t = \{V^t, E^t, X^t\}$, where $V^t$ and $E^t$ are the sets of nodes and edges in snapshot $G^t$, respectively, containing all nodes and edges that have appeared in $\{G^1, \ldots, G^t\}$. The topology of the graph $G^t$ is represented by the adjacency matrix $A^t \in \mathbb{R}^{N \times N}$, where $N$ is the number of nodes. Additionally, $X^t \in \mathbb{R}^{N \times d}$ represents the node features at time step $t$, where $d$ is the feature dimension of each node.

Our goal is to learn a parameterized model $\Phi(\cdot; \theta)$ that maps node features $x^t$ at each time step to a fixed-dimensional embedding $\mathbb{R}^d$ ($d \ll$ input feature dimension), capturing dynamic node evolution for tasks like temporal node classification.

## 4. Delay-DGSN

This section introduces the proposed Delay-DGSN in detail, as shown in Figure 2. We will introduce the method from the following three aspects: dynamic graph sampling and feature encoding, delay convolution and temporal modeling, and temporal feature aggregation and model optimization.

### 4.1. Dynamic Graph Sampling and Feature Encoding

In dynamic graph modeling based on SNNs, graph sampling and node spike feature encoding are fundamental components, also integral to Delay-DSGN. To clarify the delay mechanism in Delay-DSGN, this section briefly introduces these components. For details, see (Li et al., 2023).

For the graph sampling, the dynamic graph is divided into snapshots over fixed time intervals, with each snapshot corresponding to a time step $t$. At each time step $t$, the graph sampler performs neighbor sampling $G^t$ and $\Delta G = G^t - G^{t-1}$. For each target node $v$, $T$ sets of neighbor nodes $\{N^1, N^2, \ldots, N^t\}$ are sampled on each graph snapshot. This division method helps capture both global structural information and fine-grained temporal node information. After neighbor sampling, the feature aggregation for the sampled neighbor set $N_v^t$ is performed through a graph convolution layer, generating the hidden representation $h_v^t$ of node $v$ at time step $t$:

$$h_v^t = h_v^t W_v \oplus \mathrm{AGG}(h_u^t W_u, u \in N_v^t), \tag{1}$$

where, when $t = 0$, the initial hidden representation is the node feature $x_v$; $W_v$ and $W_u$ are learnable weight matrices; the aggregation function AGG is average aggregation; $\oplus$

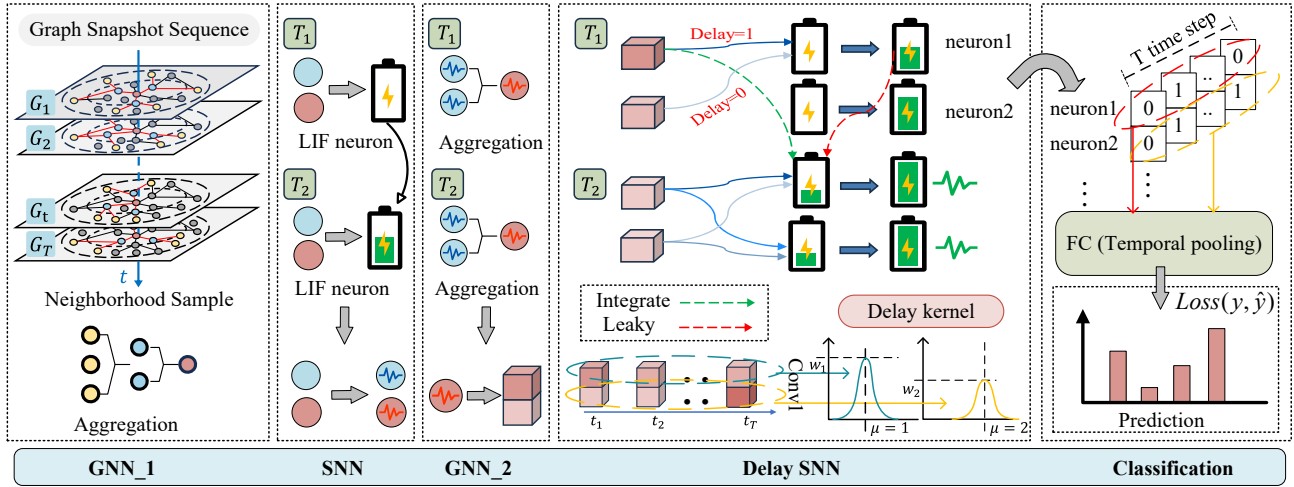

*Figure 2.* An overview of the Delay-DSGN. The dynamic graph is divided into fixed-interval snapshots. For each snapshot, second-order neighbors are sampled, and GNN aggregates neighbor information, while SNN encodes the information into spike representations. In the Delay-SNN layer, the model adaptively adjusts the center of the Gaussian delay kernel to capture varying temporal delay characteristics, generating node-level delay representations. A multi-timescale fusion mechanism further integrates representations across different time steps, ultimately producing the final embeddings for downstream tasks.

denotes vector addition.

For the node spike feature encoding, after generating the hidden representations of nodes through neighbor aggregation, the LIF (Leaky Integrate-and-Fire) neuron mechanism is utilized to convert the float-valued representation into sparse spike representation $s_v^t$. This process generates a set of spike representations for the node over $T$ time steps: $\{s_v^1, s_v^2, \ldots, s_v^T\}$. This spike encoding not only retains the essential information of the node features but also reduces computational complexity through sparsification.

### 4.2. Delay Convolution and Temporal Modeling

Delay convolution and temporal modeling are core to Delay-DGSN, capturing temporal dependencies and historical influences on node representations through delay convolution and the LIF neuron mechanism.

#### 4.2.1. DELAY CONVOLUTION KERNEL

Inspired by temporal convolution in CNNs, Delay-DGSN uses a delay convolution kernel of size $K_s$, with $K_s - 1$ as the maximum delay step. To maintain the sequence length after convolution, the node feature spike sequence $s_v^{(j,t)}$ (where $j$ is a node feature) is zero-padded on the left by $K_s - 1$ steps:

$$s_v^{(\tilde{j},t)} = [0, 0, \ldots, 0, s_v^{(j,t)}], \quad (2)$$

To implement temporal shifts, Delay-DGSN uses a Gaussian kernel function to construct the delay convolution kernel

$k_{ij}$, defined as:

$$k_{ij}[n] = w_{ij} \exp\left(\frac{-(n - (K_s - d_{ij} - 1))^2}{2\sigma^2}\right), \quad (3)$$

where $d_{ij} \in [0, K_s - 1]$ and $w_{ij}$ are the synaptic delay and weight between neurons $i$ and $j$; $\sigma \in \mathbb{R}^*$ is the standard deviation; and $n \in [0, \ldots, K_s - 1]$ denotes the temporal delay index of the kernel. The introduction of the Gaussian kernel aims to smoothly model the delay effect at different time steps. By learning $d_{ij}$, the model can dynamically adjust the center of the Gaussian kernel, accurately capturing information under different time delays. The learned delays have a certain degree of interpretability, as detailed in Appendix B. The synaptic weight $w_{ij}$ further enhances the model's ability to adaptively adjust the importance of each synapse, improving the model's expressiveness and flexibility. The delay convolution kernel $k_{ij}$ is then convolved with the padded spike sequence $s_v^{(\tilde{j},t)}$ to produce the delay feature input:

$$I_v^{(j,t)} = k_{ij} * s_v^{(\tilde{j},t)}, \quad (4)$$

This delay representation preserves current spike information while integrating historical spikes, with the delay convolution kernel weighting contributions across time, enriching the node representation with temporal dynamics.

#### 4.2.2. TEMPORAL MODELING

The delay features obtained from Equation (4) are fed into LIF neurons for dynamic membrane potential updates and

spike generation. The LIF neuron simulates the basic behavior of biological neurons, and its computation can be expressed as:

$$\text{Integrate: } V_v^{(i,t)} = \beta V_v^{(i,t-1)} + (1-\beta)\sum_j I_v^{(j,t)}, \quad (5)$$

$$\text{Fire: } S_v^{(i,t)} = \Theta(V_v^{(i,t)} - V_{th}^{(i,t-1)}), \quad (6)$$

$$\text{Rest: } V_v^{(i,t)} = V_v^{(i,t)} - S_v^{(i,t)}V_{th}^{(i,t-1)}, \quad (7)$$

$$\text{Update: } V_{th}^{(i,t)} = \gamma S_v^{(i,t)} + \tau_{th}, V_{th}^{(i,t-1)} \quad (8)$$

where $V_v^{(i,t)}$ represents the membrane potential of neuron $i$ for node $v$ at time step $t$, and $\beta$ is the leakage constant controlling the decay between the current input and historical information. $\Theta$ denotes the step function, and $V_{th}^{(i,t-1)}$ is the threshold of neuron $i$ at the previous time step. When $V_v^{(i,t)} - V_{th}^{(i,t-1)} > 0$, the membrane potential exceeds the threshold, triggering a spike, $S_v^{(i,t)} = 1$; otherwise, $S_v^{(i,t)} = 0$. We adopt a soft reset mechanism, which, compared to hard reset, retains part of the voltage exceeding the threshold, thereby preserving more information. The neuron's threshold is updated adaptively, as proposed in SpikeNet, by adjusting the decay factors $\gamma$ and $\tau_{th}$.

By combining delay convolution with the LIF neuron mechanism, Delay-DGSN generates node representations rich in temporal information. Delay convolution integrates spike information across time steps using a Gaussian kernel, while LIF neurons refine it through dynamic membrane updates. The resulting node representation $z_v^t$ incorporates both current and historical information from the past $K_s - 1$ steps, effectively modeling temporal dynamics. Learnable delay parameters $d_{ij}$ further allow adaptive adjustments, enabling Delay-DGSN to handle complex temporal dependencies in dynamic graphs with greater flexibility and expressiveness.

### 4.3. Temporal Feature Aggregation and Model Optimization

The delay convolution primarily addresses the problem of weighted aggregation of historical information within a single time step. In dynamic graphs, topological changes (such as the evolution of communities in social networks) often span multiple time steps. Therefore, aggregating information from only a single time step is insufficient to fully capture these complex topological variations. By integrating node representations across different time steps into a unified feature, it becomes possible to better model these long-term dependencies and intricate topological evolutions. In addition, this approach compensates for the information loss caused by the spiking activation characteristics in SNNs. Specifically, the model stacks the node representations $z_v^t$ across all time steps to form a feature matrix $Z_v \in \mathbb{R}^{d \times T}$ that incorporates temporal dimension information. Then, a

trainable weight matrix $W_p \in \mathbb{R}^{d \times T}$ is applied to perform weighted pooling along the temporal dimension, enabling weighted fusion of features from different time steps:

$$Z_v \cdot W_p = \sum_{t=1}^{T} Z_{i,t} \odot W_{i,t}, \quad (9)$$

where $i \in [1, d]$ denotes the $i$-th feature, and $W_{i,t}$ represents the importance of feature $i$ at different time steps. The final result $Z_v \cdot W_p \in \mathbb{R}^d$ is obtained by performing element-wise multiplication between $Z_v$ and $W_p$, followed by row-wise summation. The aggregation process fuses temporal information into a unified representation, which is fed into a fully connected layer to generate node classification results.

During optimization, Delay-DGSN uses a temporal regularization term to penalize discrepancies between feature representations at different time steps, reducing noise from insignificant temporal fluctuations. The regularization term $R$ is defined as:

$$R = \sum_{t=1}^{T} \|z_v^t - z_v^{t-1}\|^2, \quad (10)$$

where $z_v^t$ and $z_v^{t-1}$ are the node representations at time steps $t$ and $t-1$, respectively. This regularization enhances robustness by smoothing consecutive node representations, focusing on meaningful long-term trends over short-term noise. Finally, Delay-DGSN is optimized with cross-entropy loss as the objective function. The overall loss function is:

$$\mathcal{L} = \mathcal{L}_{\text{CE}} + \lambda R, \quad (11)$$

where $\mathcal{L}_{\text{CE}}$ is the cross-entropy loss and $\lambda$ is the regularization coefficient.

## 5. Theoretical Analysis

In the Delay-DGSN model, the Gaussian delay kernel is used to capture the temporal dependencies between nodes. To ensure that the model effectively controls the gradient flow during training and avoids issues such as gradient explosion and vanishing gradients, the following theorem provides the necessary conditions for the standard deviation $\sigma$ and kernel size $K_s$ of the Gaussian delay kernel.

**Theorem 5.1** (Gradient Stability and Gaussian Delay Kernel Condition). *In the Delay-DGSN model, assume the loss function $L$ is differentiable with respect to the output $h_i^t$ of feature $i$ at time step $t$, and the model uses a Gaussian delay kernel $k_{ij}[n]$ with standard deviation $\sigma$ and kernel size $K_s$. The Gaussian delay kernel has the form:*

$$k_{ij}[n] = \exp\left(-\frac{(n - \mu_{ij})^2}{2\sigma^2}\right), \quad (12)$$

*where $\mu_{ij} = K_s - d_{ij} - 1$, and $d_{ij}$ is the learnable delay parameter. Let $W_{max} = \max_{i,j}|w_{ij}|$ and $K_{max} =$*

$\max_i \sum_{j=1}^{F} |w_{ij}|$, *where $F$ is the total feature dimension. If the Gaussian delay kernel's standard deviation $\sigma$ and kernel size $K_s$ satisfy the following condition:*

$$\sigma \geq \sqrt{\frac{(K_s - 1)^2}{8 \ln\left(\frac{K_{max}}{W_{max}}\right)}}, \quad (13)$$

*then, during the backpropagation process, the gradient of the spike sequence $s_j^{(t-n)}$ with respect to the loss function $\frac{\partial L}{\partial s_j^{(t-n)}}$ satisfies the following bound for any pair of features $i$ and $j$:*

$$\left| \frac{\partial L}{\partial s_i^{(t-n)}} \right| \leq C \cdot \sum_{j=1}^{F} \left| \frac{\partial L}{\partial h_j^t} \right|, \quad (14)$$

*where the constant $C$ is given by:*

$$C = \sum_{j=1}^{F} |w_{ij}| \cdot \exp\left(-\frac{(K_s - 1 - n' - \mu_{ij})^2}{2\sigma^2}\right), \quad (15)$$

*Proof.* The proof of this theorem involves deriving the gradient bounds under the assumption of Gaussian delay kernel behavior and utilizing the properties of temporal spike sequences in the Delay-DGSN model. Full details can be found in the Appendix A.

Under the given conditions, this theorem ensures Delay-DGSN effectively controls gradient propagation, preventing explosion or vanishing. It also guides parameter selection for the Gaussian delay kernel, enabling accurate temporal modeling and stable training.

# 6. Experiments

## 6.1. Datasets

To evaluate the effectiveness of Delay-DSGN, we selected three large-scale datasets, which are commonly used for validating dynamic graph representation learning models. These datasets are DBLP (Lu et al., 2019), Tmall (Lu et al., 2019), and Patent (Hall et al., 2001), as listed in Table 1.

*Table 1.* The statistics of datasets.

|  | DBLP | Tmall | Patent |
|---|---|---|---|
| Nodes | 28,085 | 577,314 | 2,738,012 |
| Edges | 236,894 | 4,807,545 | 13,960,811 |
| Time steps | 27 | 186 | 25 |
| Classes | 10 | 5 | 6 |

The DBLP, from a computer science research database, represents authors as nodes, with edges indicating collaborations, and papers classified into 10 categories. The Tmall,

based on Tmall sales records, is a bipartite user-product graph with edges representing purchases, and the top five product categories as labels. The Patent, from the U.S. patent database, has nodes for patents, edges for citations, and six main categories. For Tmall and Patent, time windows of 10 and 2 generate 19 and 13 graph snapshots, respectively. Table 1 summarizes the dataset statistics.

## 6.2. Evaluation Metrics

To comprehensively evaluate the classification performance of the model, we use two commonly used metrics: F1-macro and F1-micro. F1-macro calculates the average of precision and recall across all classes, making it suitable for imbalanced datasets. Its formula is:

$$F1_{\text{macro}} = \frac{1}{n} \sum_{i=1}^{n} \frac{2P_i R_i}{P_i + R_i}, \quad (16)$$

where $P_i$ and $R_i$ represent the precision and recall of class $i$, and $n$ is the total number of classes.

F1-micro, on the other hand, computes the overall precision and recall across all classes, making it suitable for balanced datasets. Its formula is:

$$F1_{\text{micro}} = \frac{2P_{\text{micro}} R_{\text{micro}}}{P_{\text{micro}} + R_{\text{micro}}}, \quad (17)$$

where $P_{\text{micro}}$ and $R_{\text{micro}}$ are the aggregated precision and recall across all classes.

## 6.3. Experimental Setup

To demonstrate the superiority of the Delay-DSGN, we compare it with 8 existing methods. These include static graph-based methods DeepWalk (Perozzi et al., 2014) and Node2Vec (Grover & Leskovec, 2016), dynamic graph methods based on point processes M2DNE (Lu et al., 2019) and HTNE (Zuo et al., 2018), RNN-based methods EvolveGCN (Pareja et al., 2020), attention-based methods TGAT (Xu et al., 2020), and two spiking methods SpikeNet (Li et al., 2023), Dy-SIGN (Yin et al., 2024b).

The experimental follows the same settings with (Li et al., 2023). For each training ratio (i.e., 40%, 60%, and 80%), we compute the F1-macro and F1-micro scores. The hidden dimension for all methods is set to 128, the batch size is 1024, and the total number of training epochs is 100.

## 6.4. Experimental Results

### 6.4.1. PERFORMANCE COMPARISON

Table 2 summarizes the classification results on the DBLP, Tmall, and Patent across various training ratios. Delay-DSGN achieves state-of-the-art performance in dynamic graph representation learning, consistently outperforming

*Table 2.* Results of node classification tasks. The results are averages from five runs with different random seeds, with the best results highlighted in bold. (Tr.ratio: training ratio, - denotes time-consuming.)

| | Metric | Tr.ratio | DeepWalk | Node2Vec | HTNE | M$^2$DNE | EvolveGCN | TGAT | SpikeNet | Dy-SIGN | Delay-SGN |
|---|---|---|---|---|---|---|---|---|---|---|---|
| **DBLP** | Ma-F1 | 40% | 67.08 | 66.07 | 67.68 | 69.02 | 67.22 | 71.18 | 70.88 | 70.94±0.1 | **72.32±0.4** |
| | | 60% | 67.17 | 66.81 | 68.24 | 69.48 | 69.78 | 71.74 | 71.98 | 72.07±0.1 | **74.16±0.3** |
| | | 80% | 67.12 | 66.93 | 68.36 | 69.75 | 71.20 | 72.15 | 74.65 | 74.67±0.5 | **76.54±0.4** |
| | Mi-F1 | 40% | 66.53 | 66.80 | 68.53 | 69.23 | 69.12 | 71.10 | 71.98 | 71.90±0.1 | **72.56±0.2** |
| | | 60% | 66.89 | 67.37 | 68.57 | 69.47 | 70.43 | 71.85 | 72.35 | 72.61±0.4 | **74.44±0.3** |
| | | 80% | 66.38 | 67.31 | 68.79 | 69.71 | 71.32 | 73.12 | 74.86 | 74.96±0.2 | **76.87±0.5** |
| **Tmall** | Ma-F1 | 40% | 67.08 | 54.37 | 54.81 | 57.75 | 53.02 | 56.90 | 58.84 | 57.48±0.1 | **60.25±0.1** |
| | | 60% | 67.17 | 54.55 | 54.89 | 57.99 | 54.99 | 57.61 | 61.13 | 60.94±0.2 | **62.56±0.3** |
| | | 80% | 67.12 | 54.58 | 54.93 | 58.47 | 55.78 | 58.01 | 62.40 | 61.89±0.1 | **64.02±0.2** |
| | Mi-F1 | 40% | 66.53 | 60.41 | 62.53 | 64.21 | 59.96 | 62.05 | 63.52 | 62.93±0.3 | **64.32±0.1** |
| | | 60% | 66.89 | 60.56 | 62.59 | 64.38 | 61.19 | 62.92 | 64.84 | 64.10±0.3 | **66.20±0.2** |
| | | 80% | 66.38 | 60.66 | 62.64 | 64.65 | 61.77 | 63.32 | 66.10 | 65.82±0.2 | **67.88±0.4** |
| **Patent** | Ma-F1 | 40% | 67.08 | 69.01 | - | - | 79.67 | 81.51 | 83.53 | 83.57±0.3 | **83.72±0.1** |
| | | 60% | 67.17 | 69.08 | - | - | 79.76 | 81.56 | 83.85 | 83.77±0.2 | **84.01±0.1** |
| | | 80% | 67.12 | 68.99 | - | - | 80.13 | 81.57 | 83.90 | 83.91±0.2 | **84.20±0.1** |
| | Mi-F1 | 40% | 66.53 | 68.14 | - | - | 79.39 | 80.79 | 83.48 | 83.50±0.2 | **83.66±0.1** |
| | | 60% | 66.89 | 68.20 | - | - | 79.75 | 80.81 | 83.80 | 83.47±0.1 | **83.97±0.1** |
| | | 80% | 66.38 | 68.10 | - | - | 80.01 | 80.93 | 83.88 | 83.90±0.2 | **84.15±0.1** |

all baseline methods. For instance, on the DBLP, it achieves an Mi-F1 score of 76.87% at an 80% training ratio, outperforming the second-best method, Dy-SIGN, by 2.21%. On the Tmall, Delay-DSGN achieves an Mi-F1 score of 67.88%, 4.56% higher than TGAT. On the large-scale Patent, it delivers the best Ma-F1 score of 84.20%, showcasing its scalability and robustness.

While dynamic methods like HTNE and M$^2$DNE improve upon static approaches by incorporating temporal information, their high computational complexity limits their scalability. As shown in Table 2, these methods fail to produce results on the large Patent due to excessive training costs. In contrast, Delay-DSGN leverages sparse binary representations and an efficient synaptic delay mechanism, enabling it to model temporal dependencies accurately while maintaining computational efficiency, making it well-suited for large-scale, real-world dynamic graphs.

### 6.4.2. ABLATION STUDY

To further investigate the impact of the delay mechanism in Delay-DSGN and evaluate the effectiveness of spike-delay learning for dynamic graph representation, we designed the following ablation experiments. We compared Delay-DSGN with two baseline models: the fixed random delay model and the no-delay model. The fixed random delay model initializes the delay values randomly at the beginning of training and keeps them constant throughout. The no-delay model removes the delay module and adjusts the hidden layer structure by increasing the number of neurons to ensure that all models have the same parameter count and

start training from the same weight initialization. Each condition was run five times, and the average result is reported.

The results in Figure 3 show that the Delay-DSGN model with the delay module performs better across all datasets, validating the importance of the delay mechanism in modeling long-term dependencies in dynamic graphs.

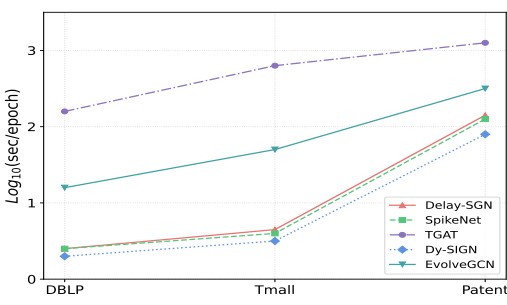

*Figure 4.* Comparison of training times for different methods on the DBLP, Tmall, and Patent. The training durations are presented in units of $log_{10}$(seconds/epoch).

### 6.4.3. TIME EFFICIENCY

To validate the computational efficiency of SNN-based methods, we compared the training time of Delay-DSGN with other dynamic graph methods (Dy-SIGN, SpikeNet, EvolveGCN, and TGAT). All experiments were run on a Titan RTX GPU, and we compared the time cost per training epoch. As shown in Figure 4, the SNN-based methods significantly reduce training time. By using sparse binary

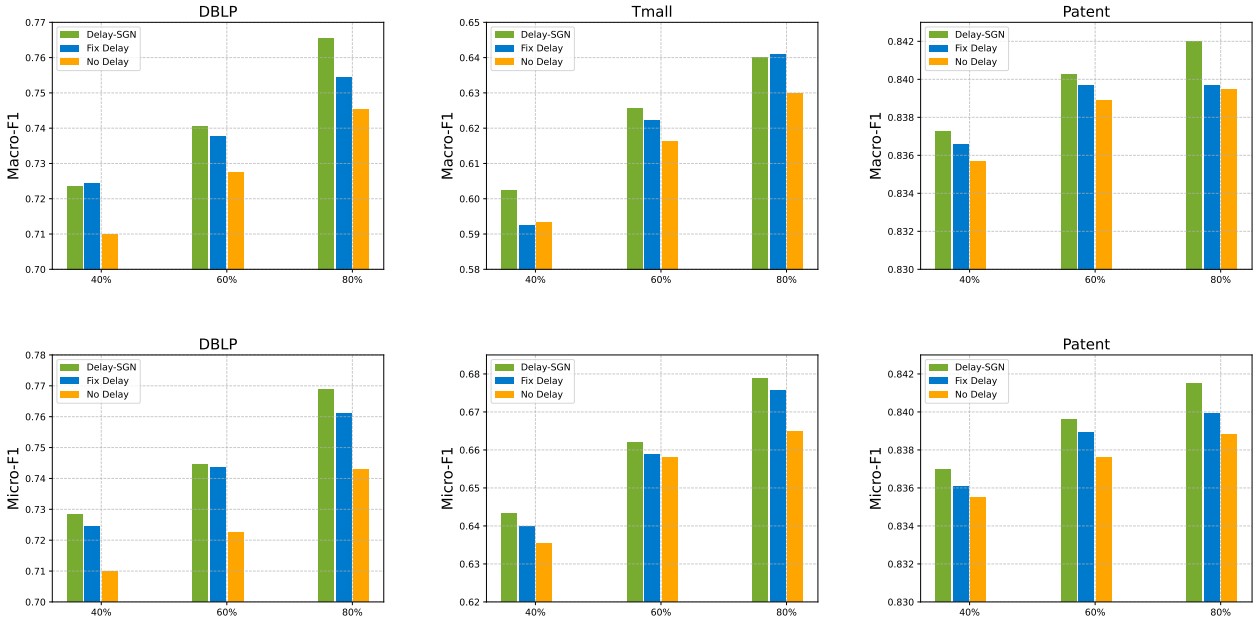

*Figure 3.* Results of the delay mechanism ablation study. The six figures respectively show the classification performance of Delay-DSGN compared to other delay mechanisms (fixed random delay, no delay) on the DBLP, Tmall, and Patent under different training set proportions.

activation functions instead of continuous real-valued activations in traditional GNNs, SNNs effectively reduce computational complexity. As a result, SNN-based methods have a clear advantage in terms of computational efficiency. Although Delay-DSGN introduces the delay module, which increases the number of parameters, this has a negligible impact on the overall training time. Compared to other SNN methods such as SpikeNet and Dy-SIGN, Delay-DSGN maintains comparable training times while achieving superior classification performance. These results demonstrate that Delay-DSGN maintains the computational efficiency of SNNs while providing significant performance improvements.

### 6.4.4. PARAMETER SENSITIVITY ANALYSIS

This section explores the effect of the maximum delay time $d_m$ and standard deviation $\sigma$ on the performance of the Delay-DSGN model. In the experiments, we set $d_m$ to $\{1, 3, 5, 7, 9\}$ and $\sigma$ to $\{0.25, 0.5, 0.75, 1, 2\}$, while keeping other parameters fixed.

The results in Figure 6 show that when $d_m = 1$, the model performs similarly to the no-delay model, as the Gaussian kernel has minimal influence. As $d_m$ increases, if $\sigma$ is set too small, the model overly focuses on the spike information at delayed time steps, causing information loss. Conversely, if $\sigma$ is too large, the weights of the delay kernel become too smooth, leading the model to assign almost equal importance to information across all time steps, losing its selective

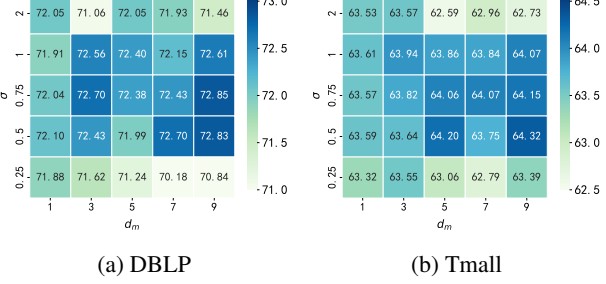

     (a) DBLP            (b) Tmall

*Figure 5.* Parameter experiment results on the DBLP and Tmall under different settings of $d_m$ and $\sigma$. In the heatmap, the color bar represents the performance score, with darker colors indicating higher performance.

memory ability. Therefore, a moderate value of $\sigma$ can retain important time step information while considering contributions from neighboring time steps. Larger $d_m$ values increase the receptive field of the delay kernel, allowing the model to better capture multi-step temporal dependencies in dynamic graphs. Experimental results on the DBLP and Tmall confirm that larger $d_m$ and moderate $\sigma$ settings significantly improve model performance.

## 7. Conclusion and Future Work

This paper proposes the Delay-DSGN, which introduces a learnable delay mechanism inspired by biological synaptic

plasticity. This approach successfully enhances the temporal correlation of information propagation between nodes and preserves historical information, effectively overcoming the challenges faced by existing dynamic graph representation methods that fail to model information propagation delays. The experimental results demonstrate that Delay-DSGN outperforms eight state-of-the-art methods across multiple large-scale dynamic graph datasets in node classification tasks. More importantly, we provide theoretical conditions for selecting the standard deviation and kernel size of the Gaussian delay kernel, ensuring that the model avoids gradient explosion and vanishing issues during training. Future work will focus on further exploring the delay mechanism to improve the model's adaptability and performance on more complex and diverse dynamic graph datasets.

## Acknowledgement

This work is supported by the National Natural Science Foundation of China (Nos. 62272285, 62376142, U21A20473).

## Impact Statement

This paper presents work whose goal is to advance the field of Machine Learning. There are many potential societal consequences of our work, none which we feel must be specifically highlighted here.

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

# A. Proof of Theorem 5.1

## A.1. Notation

- $L$: Loss function.

- $h_j^t$: Output of feature $j$ at time step $t$.

- $s_i^{(t-n)}$: Spike firing sequence of feature $i$ at time step $t - n$.

- $k_{ij}[n]$: Gaussian delay kernel value from feature $i$ to $j$, with delay steps $K_s - 1 - n$.

- $w_{ij}$: Synaptic weight from feature $i$ to feature $j$.

- $\sigma$: Standard deviation of the Gaussian delay kernel, controlling the diffusion of delayed signals.

- $K_s$: Size of the Gaussian delay kernel, with $K_s - 1$ indicating the maximum delay steps.

- $d_{ij}$: Delay steps between features $i$ and $j$, a learnable parameter in the model.

- $K_{\max}$: Maximum value of the sum of absolute synaptic weights for any feature $i$, i.e., $K_{\max} = \max_i \sum_{j=1}^{F} |w_{ij}|$.

- $W_{\max}$: Maximum absolute synaptic weight, i.e., $W_{\max} = \max_{(i,j)} |w_{ij}|$.

- $F$: Total number of features.

- $C$: Upper bound constant for gradient propagation, defined as the sum of the product of synaptic weights and Gaussian delay kernel values.

## A.2. Proof Process

### Step 1: Apply the Chain Rule to Expand the Gradient

By the chain rule, the gradient of the loss function $L$ with respect to the spike firing sequence $s_i^{(t-n)}$ is expressed as:

$$\frac{\partial L}{\partial s_i^{(t-n)}} = \sum_{j \in F} \frac{\partial L}{\partial h_j^t} \cdot \frac{\partial h_j^t}{\partial s_i^{(t-n)}}, \tag{18}$$

where:

- $\frac{\partial L}{\partial h_j^t}$ is the gradient of the loss function with respect to the output of feature $j$ at time step $t$.

- $\frac{\partial h_j^t}{\partial s_i^{(t-n)}}$ represents the impact of a spike fired by feature $i$ at time step $t - n$ on the output of feature $j$ at time step $t$.

### Step 2: Calculate $\frac{\partial h_j^t}{\partial s_i^{(t-n')}}$

In the Delay-DSGN model, the output $h_j^t$ of feature $j$ at time step $t$ is defined as:

$$h_j^t = \sum_{i \in F} \sum_{n=0}^{K_s-1} w_{ij} \cdot k_{ij}[K_s - 1 - n] \cdot s_i^{(t-n)}, \tag{19}$$

Thus, for a specific $n = n'$, the impact of the spike fired by feature $i$ at time step $t - n'$ on the output of feature $j$ at time step $t$ is:

$$\frac{\partial h_j^t}{\partial s_i^{(t-n')}} = \sum_{j \in F} w_{ij} \cdot k_{ij}[K_s - 1 - n'], \tag{20}$$

where the Gaussian delay kernel $k_{ij}[n]$ is defined as:

$$k_{ij}[n] = \exp\left(\frac{-(n - \mu_{ij})^2}{2\sigma^2}\right), \quad \mu_{ij} = K_s - d_{ij} - 1, \tag{21}$$

**Step 3: Bound the Gradient**

Substituting the above results into the gradient expression, we get:

$$\left|\frac{\partial L}{\partial s_i^{(t-n)}}\right| = \left|\sum_{j=1}^{F} \frac{\partial L}{\partial h_j^t} \cdot \sum_{j=1}^{F} w_{ij} \cdot k_{ij}[K_s - 1 - n']\right|, \tag{22}$$

Using the triangle inequality and the properties of absolute values, we obtain:

$$\left|\frac{\partial L}{\partial s_i^{(t-n)}}\right| \le \sum_{j=1}^{F} \left|\frac{\partial L}{\partial h_j^t}\right| \cdot \sum_{j=1}^{F} |w_{ij}| \cdot k_{ij}[K_s - 1 - n'], \tag{23}$$

Define the constant $C$ as:

$$C = \sum_{j=1}^{F} |w_{ij}| \cdot k_{ij}[K_s - 1 - n'] = \sum_{j=1}^{F} |w_{ij}| \cdot \exp\left(\frac{-(K_s - 1 - n' - \mu_{ij})^2}{2\sigma^2}\right), \tag{24}$$

Thus, the gradient is bounded by:

$$\left|\frac{\partial L}{\partial s_i^{(t-n)}}\right| \le C \cdot \sum_{j=1}^{F} \left|\frac{\partial L}{\partial h_j^t}\right|, \tag{25}$$

**Step 4: Ensure $C$ is Bounded to Prevent Gradient Explosion and Vanishing**

To prevent $C$ from becoming too large or too small, and avoid gradient explosion or vanishing, we impose appropriate constraints on the Gaussian delay kernel's standard deviation $\sigma$ and kernel size $K_s$.

*Preventing Gradient Explosion:*

Since $W_{\max} = \max_{(i,j)} |w_{ij}|$ and $K_{\max} = \max_i \sum_{j=1}^{F} |w_{ij}|$, and also $\sum_{j=1}^{F} |w_{ij}| \le K_{\max}$, we have:

$$C = \sum_{j=1}^{F} |w_{ij}| \cdot \exp\left(\frac{-(K_s - 1 - n' - \mu_{ij})^2}{2\sigma^2}\right) \le \sum_{j=1}^{F} |w_{ij}| \le K_{\max}, \tag{26}$$

This guarantees that $C$ will not exceed $K_{\max}$, preventing gradient explosion.

*Preventing Gradient Vanishing:*

To prevent $C$ from being too small, we ensure that the value of the Gaussian delay kernel at the boundaries of the delay range (i.e., $n = 0$ and $n = K_s - 1$) is above a threshold $\epsilon$, i.e.,

$$\exp\left(\frac{-(n - \mu_{ij})^2}{2\sigma^2}\right) \ge \epsilon, \tag{27}$$

Assuming the delay center $\mu_{ij}$ is at the center of the kernel, i.e., $\mu_{ij} = \frac{K_s-1}{2}$, the maximum distance between $n$ and $\mu_{ij}$ is:

$$|n - \mu_{ij}| \leq \frac{K_s - 1}{2}, \tag{28}$$

Thus, to ensure that the Gaussian delay kernel's value does not fall below $\epsilon$ at the boundaries $n = 0$ and $n = K_s - 1$, we require:

$$\exp\left(\frac{-\left(\frac{K_s - 1}{2}\right)^2}{2\sigma^2}\right) \geq \epsilon, \tag{29}$$

Taking the natural logarithm of both sides:

$$\frac{-(K_s - 1)^2}{8\sigma^2} \geq \ln(\epsilon), \tag{30}$$

Solving for $\sigma$:

$$\sigma^2 \geq \frac{(K_s - 1)^2}{8(-\ln(\epsilon))}, \tag{31}$$

Thus,

$$\sigma \geq \sqrt{\frac{(K_s - 1)^2}{8(-\ln(\epsilon))}}, \tag{32}$$

To ensure that $\epsilon$ accounts for the influence of all feature dimensions, we set:

$$\epsilon = \frac{K_{\max}}{W_{\max}}, \tag{33}$$

Therefore:

$$\sigma \geq \sqrt{\frac{(K_s - 1)^2}{8(-\ln(\epsilon))}} = \sqrt{\frac{(K_s - 1)^2}{8\ln\left(\frac{W_{\max}}{K_{\max}}\right)}}, \tag{34}$$

This ensures that the standard deviation $\sigma$ of the Gaussian delay kernel is sufficiently large to prevent the kernel values from becoming too sharp, thus avoiding the gradient vanishing issue.

## B. Interpretability of Delay

In our model, the delay parameters are initialized randomly following a normal distribution within a predefined range. This initialization strategy ensures that the model begins with a diverse set of potential delay values, allowing it to explore various temporal dependencies during training. For instance, when the maximum delay is constrained to 5 time units, the initial delay values are sampled from a normal distribution defined over the interval $[0, 5]$.

After training on the DBLP, we observe that the learned delay distribution slightly shifts to the right compared to the initial distribution. This shift indicates that in academic settings, interactions and influences often take longer to materialize, reflecting the typically slower pace at which scholarly collaborations develop and propagate through the network. In contrast, when evaluating the model on the Tmall, the learned delay distribution exhibits a slight leftward skew. This behavior aligns well with the nature of online consumer behavior, where users tend to respond quickly to items or promotions, resulting in shorter interaction delays. The Patent exhibits larger delays, with a more uniform distribution toward the right, suggesting that the impact of patents generally takes a longer time to diffuse and manifest. Despite sharing the same initialization

distribution, the model consistently learns dataset-specific delay values across multiple training runs, demonstrating its ability to adapt to inherent temporal characteristics.

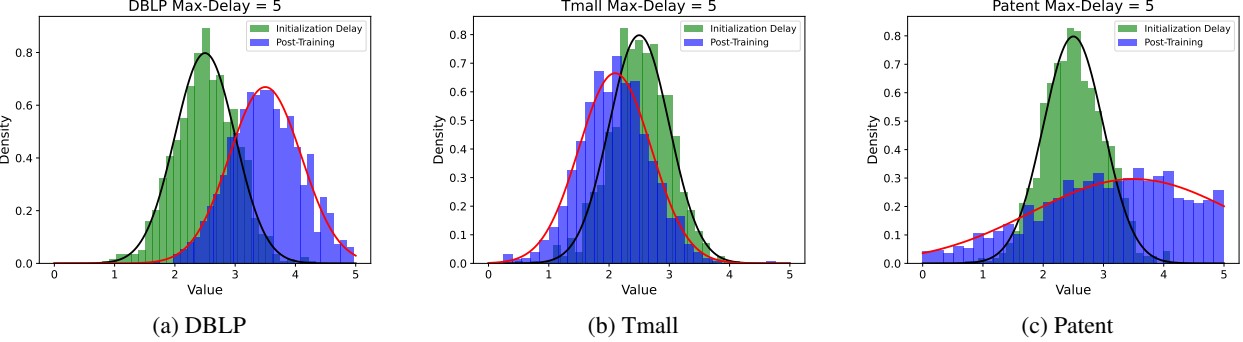

(a) DBLP        (b) Tmall        (c) Patent

*Figure 6.* The probability density histograms of the initial and trained models across three datasets.

