# OpenReview forum: "Delay-DSGN: A Dynamic Spiking Graph Neural Network with Delay Mechanisms for Evolving Graph"
_ICML.cc/2025/Conference — ICML 2025 poster_

### Official Review · Reviewer_WWHS · 2025-03-09

**Overall Recommendation:** 4

**Summary:**

This paper introduces Delay-DSGN, a novel dynamic spiking graph neural network that incorporates a learnable synaptic delay mechanism to enhance dynamic graph representation learning. The authors argue that existing SNN-based methods struggle with temporal information propagation and historical forgetting, and they propose a Gaussian delay kernel to address this issue. The model is evaluated on three large-scale dynamic graph datasets for node classification, demonstrating superior performance over eight state-of-the-art baselines. The authors also provide a theoretical analysis of the stability conditions for the delay kernel, ensuring the model avoids gradient explosion and vanishing issues.

**Claims And Evidence:**

The paper makes several key claims:
1. Delay mechanisms improve long-term temporal modeling in dynamic graphs.
2. Delay-DSGN outperforms existing dynamic graph models (e.g., Dy-SIGN, SpikeNet) in node classification.
3. Theoretical analysis ensures stable training of the Gaussian delay kernel.

The evidence provided supports these claims reasonably well:
1. The experimental results on three large-scale datasets demonstrate consistent improvements over state-of-the-art methods.
2. The ablation study comparing learnable delays vs. fixed delays vs. no delay strengthens the argument that delayed information propagation is beneficial.
3. The theoretical section derives conditions for stable training, although additional empirical validation (e.g., loss landscape analysis) would further reinforce this claim.

However, the paper does not provide enough analysis on the interpretability of learned delays—are the learned delays consistent across different datasets?

**Essential References Not Discussed:**

The paper should consider discussing:
1. “Temporal Graph Networks” (Rossi et al., 2020, ICLR), which also models long-term dependencies in dynamic graphs using memory-based approaches.
2. “A Hawkes process-based graph neural network: TREND” (Wen et al., 2022, WWW), which provides an alternative view on modeling temporal dependencies.

**Experimental Designs Or Analyses:**

The authors claim that the delay mechanism effectively preserves historical information and mitigates the information forgetting issue in dynamic graphs. This claim is supported by comprehensive experimental results showing improvements in both macro and micro F1 scores, as well as a theoretical gradient stability proof provided in the appendix. The evidence is generally convincing, although some parameter choices could benefit from clearer justification.

**Methods And Evaluation Criteria:**

The choice of node classification as the primary evaluation task and the focus on classification accuracy (Ma-F1, Mi-F1) is appropriate, as it aligns with common benchmarks in dynamic graph learning. However, the dataset-specific hyperparameter tuning is not fully discussed.

**Other Comments Or Suggestions:**

1. Typos:
I. Page 2, Line 15: "effectively mitigating historical information forgetting" → "effectively mitigating the forgetting of historical information".
II. Page 4, Equation (5): Should reference Equation (4) for clarity.
2. The paper mentions a comparison with 8 methods, but the experimental results only present 7. Is this due to an omission of one method or a typographical error?

**Other Strengths And Weaknesses:**

Strengths:
1. A novel delay-based temporal mechanism, inspired by biological synaptic plasticity, which offers a fresh perspective in GNN research.
2. Impressive empirical results on large-scale dynamic graphs, with meaningful comparisons to both non-spiking and spiking methods.
3. A solid mathematical foundation, providing a theoretical basis for stable training.

Weaknesses:
The interpretability of learned delays is somewhat limited—it’s not entirely clear what the model is learning and whether the delays correspond to real-world temporal phenomena.

**Questions For Authors:**

1. Are the learned delays consistent across training runs, or do they vary significantly with different initializations?
2. How does Delay-DSGN compare to attention-based methods like TGN in terms of handling long-term dependencies?

**Relation To Broader Scientific Literature:**

This work aligns well with research in dynamic graph learning and spiking neural networks, building on prior studies in GNNs for dynamic graphs (TGAT, EvolveGCN, Dy-SIGN) and spiking neural networks for graphs (SpikeNet).

**Theoretical Claims:**

The paper provides a sound theoretical analysis of the conditions required for stable training of the Gaussian delay kernel, ensuring that the model avoids gradient explosion and vanishing issues. The proof (Appendix A) correctly applies the chain rule and leverages the properties of Gaussian functions to derive a well-structured bound on the standard deviation ($\sigma$) and kernel size ($K_s$).

The derivation is mathematically reasonable, and the constraints appear well-motivated. One potential improvement would be an empirical validation of how training behaves when $\sigma$ is set outside the derived bounds. A brief numerical analysis confirming the theoretical stability conditions would further strengthen this claim.

---

> ### Author Rebuttal · Authors · 2025-03-31
>
> Thank you for your meticulous review of our work and the valuable feedback provided.
>
> **Response to Weaknesses and Question 1:**
>
> In our model, delay parameters are randomly initialized following a normal distribution within a specified range. For example, when the maximum delay is set to 5, the initial delays are sampled from a normal distribution over the interval $[0, 5]$. Training results on the DBLP show that the learned delay distribution is slightly shifted to the right, indicating that academic collaborations and influences often require a longer time to become evident. For the Tmall, the learned delay distribution is slightly skewed to the left, reflecting the typically rapid user responses and interactions in e-commerce environments. The Patent exhibits larger delays, with a more uniform distribution toward the right, suggesting that the impact of patents generally takes a longer time to diffuse and manifest. Despite sharing the same initialization distribution, the model consistently learns dataset-specific delay values across multiple training runs, demonstrating its ability to adapt to inherent temporal characteristics.
> | Delay Interval | Initialization Density | DBLP Post-Training Density | Tmall Post-Training Density | Patent Post-Training Density |
> |----------------|------------------------|----------------------------|-----------------------------|------------------------------|
> | 0-1           | 0.0  | 0.0| 0.08  | 0.0|
> | 1-2           | 0.2       | 0.17| 0.29         | 0.13                         |
> | 2-3           | 0.6     | 0.5                        | 0.46                        | 0.35                         |
> | 3-4           | 0.2                    | 0.28                       | 0.17                        | 0.33                         |
> | 4-5           | 0.0                    | 0.05                       | 0.0                           | 0.19                         |
>
> **Response to Theoretical Claims:**
>
> Thank you for your valuable feedback. We have added an extra analysis in the experimental section and visualized the training loss curves. Unfortunately, we are unable to display the figures here, so we provide the following explanation. When $σ$ is outside the derived range, the model's loss during training on the DBLP remains largely unchanged. In contrast, when $σ$ is within the derived range, the training loss consistently decreases, indicating that the model is able to learn and optimize effectively.
>
> **Response to Essential References Not Discussed and Question 2:**
>
> We have added a new paragraph in the related work section as follows:
> "Unlike the memory module in TGN, our delay mechanism explicitly models multi-step dependencies through differentiable temporal kernels, thereby mitigating the gradient decay issues inherent in RNN architectures. Compared to the Hawkes process used in TREND, the biologically inspired spike-delay mechanism in Delay-DSGN is better suited for handling discretized timestep-based graph updates. " We have compared our approach with attention-based methods such as TGAT. We will endeavor to include experimental comparisons with TGN to further demonstrate the effectiveness and advantages of our proposed method.
>
> **Response to Other Comments Or Suggestions:**
>
> Thank you for your valuable feedback and careful corrections on our paper. Based on your suggestions, we have carefully revised and improved the issues mentioned in the text as follows:
> * In line 15 on page 2, “effectively mitigating historical information forgetting” has been revised to “effectively mitigating the forgetting of historical information” for improved clarity and grammatical accuracy.
> * At the location of Equation (5) on page 4, we have added a reference to Equation (4) to better clarify the relationships between the different parts of the formulas.
> * Regarding the comparison with eight methods but only seven results being presented in the experiments, we confirmed that this was due to the omission of one method in the table. We have now updated the table to include all the mentioned methods. Due to character limitations, the results of the omitted comparison methods on the three datasets are as follows:
>
> | Dataset | Metrics | Training | DeepWalk |
> |---------|---------|----------|----------|
> | DBLP    | Ma-F1 | 40%| 67.08  |
> |         |         | 60%| 67.17|
> |         |         | 80%      | 67.12    |
> |         | Mi-F1   | 40%| 66.53    |
> |         |         | 60%| 66.89    |
> |         |         | 80%| 66.38    |
> | Tmall   | Ma-F1   | 40%| 49.09    |
> |         |         | 60%| 49.29    |
> |         |         | 80%  | 49.53    |
> |         | Mi-F1   | 40%| 57.11    |
> |         |         | 60%| 57.34    |
> |         |         | 80%| 57.88    |
> | Patent  | Ma-F1   | 40%| 72.32    |
> |         |         | 60%| 72.25    |
> |         |         | 80%  | 72.05    |
> |         | Mi-F1   | 40% | 71.57    |
> |         |         | 60% | 71.53    |
> |         |         | 80%| 71.38    |

---

### Official Review · Reviewer_nXzL · 2025-03-11

**Overall Recommendation:** 4

**Summary:**

This paper introduces Delay-DSGN, a dynamic spiking graph neural network that incorporates a learnable delay mechanism to enhance the representation of evolving graphs. By modeling synaptic delays with a Gaussian kernel, the model effectively captures temporal dependencies and mitigates information forgetting. The paper provides theoretical guarantees to address gradient issues and demonstrate the model's effectiveness through experiments on three large-scale dynamic graph datasets.

Overall, the paper offers a timely and promising contribution to dynamic graph learning and spiking neural networks.

**Claims And Evidence:**

This claim is supported by comprehensive experimental results showing improvements in both macro and micro $F1$ scores, as well as a theoretical gradient stability proof provided in the appendix.

The evidence is generally convincing, although some parameter choices could benefit from clearer justification.

**Essential References Not Discussed:**

The paper lacks a discussion of some relevant temporal graph learning works, such as GC-LSTM, a model with an LSTM embedded in a GCN, and DyRep, which uses a temporal point process to model event-driven graph evolution.

A brief discussion of these methods, particularly how they differ from Delay-DSGN, would enhance the paper’s contextual positioning.

**Experimental Designs Or Analyses:**

>The experiments are well-structured and reproducible, but a few aspects could be improved:

•	The paper reports standard deviations, but statistical significance tests (e.g., t-tests) are not provided to confirm that the improvements are meaningful rather than random variation.

•	While the impact of $\sigma$ and $d_m$ is explored (Figure 5), the authors do not discuss the computational cost of larger delay windows.

**Methods And Evaluation Criteria:**

The use of F1-macro and F1-micro as evaluation metrics is appropriate for the node classification tasks presented.

The method details are mostly clear; however, some parts of the zero-padding and convolution process could be described in greater detail for complete reproducibility.

**Other Comments Or Suggestions:**

The network architecture is shown in Figure 2, but it lacks a more detailed explanation.

The integration of Spiking Neural Networks with the graph neural network approach should be explained more clearly.

How does the SNN layer interact with the graph convolution layer, and what benefits does this bring to dynamic graph representation learning?

**Other Strengths And Weaknesses:**

>Strengths:

•	The incorporation of a learnable delay mechanism in the spiking graph neural network framework is a novel idea, blending insights from neuroscience and graph learning.

•	The paper presents theoretical guarantees for the stability of the training process, ensuring the model avoids gradient explosion or vanishing, which is an important contribution.

>Weaknesses:

•	The authors did not present the learning process or distribution characteristics of the delay mechanism in the graph data. Relevant evidence would provide a clearer explanation.

**Questions For Authors:**

1. Does the delay mechanism perform differently on irregular graphs, such as those with sparse or heterogeneous structures?

2. Additionally, the paper adopts a Gaussian delay kernel—what was the rationale behind this choice?'



## update after rebuttal

Thank you for the detailed response, which effectively addressed my main concerns. I appreciate the clarity provided and am happy to update my score accordingly. I recommend Accept.

**Relation To Broader Scientific Literature:**

The paper is well-positioned within dynamic graph learning and spiking neural network literature.

However, while the discussion on the biological plausibility of delay mechanisms is interesting, it could benefit from more references to neuromorphic computing models (e.g., event-driven SNNs).

Additionally, there is limited comparison between delay-based mechanisms and other temporal modeling approaches, such as continuous-time point processes used in DyRep and TGN.

**Theoretical Claims:**

The proof assumes that the Gaussian kernel’s influence is stable across different graph structures, but real-world dynamic graphs may have highly irregular temporal patterns. $Relevant$ $discussion$ on the $robustness$ of the theoretical bounds in these settings is necessary.

---

> ### Author Rebuttal · Authors · 2025-03-31
>
> Thank you for your meticulous review of our work and the valuable feedback provided.
>
> **Response to Experimental Designs or Analyses 1:**
>
> In our experiments, we compared eight methods across three datasets using two metrics. To assess the statistical significance of improvements by our proposed method over the baselines, we conducted Friedman tests. The results are:
> |Metric |F-Value |P-Value|
> |-|-|-|
> |Ma-F1|18.1170|0.0204|
> |Mi-F1|18.6072|0.0171|
>
> With all P-values are less than the significance level $ \alpha = 0.05 $, indicating that our method's average ranking is significantly different from the baselines for both metrics.
>
> **Response to Essential References Not Discussed:**
>
> We have made additions to the "Related Work" section: "GC-LSTM integrates LSTM into GCN to model the temporal evolution of node features. It captures temporal dependencies through hidden state transitions. In contrast, Delay-DSGN introduces a delay convolution kernel to explicitly parameterize propagation delays. DyRep utilizes temporal point processes to model the triggering intensity and time intervals of events. Its core lies in describing the probabilistic patterns of event occurrences rather than capturing the dynamic delay effects of information propagation."
>
> **Response to Weaknesses:**
>
> Please refer to the first response to Reviewer WWHS.
>
> **Response to Concerns:**
> * **Regarding zero-padding, convolution processes:** For example, at the current timestep, two neurons from the previous layer aggregate information to one neuron in the current layer. The inputs are spike trains from these two neurons, each containing activity information across four timesteps (the current timestep and three historical timesteps). Neurons exhibit different delays. Before convolution, the input sequences undergo left-side zero-padding (with padding length equal to $K_s−1$) to accommodate the convolution kernel size. Convolution kernel construction: The hyperparameter $K_s$ represents the size of the delay convolution kernel, and $K_s−1$ is the maximum delay time. The values in the delay convolution kernel follow a Gaussian distribution, with the center position located at $K_s−d−1$. Subsequently, the padded sequences are convolved with the delay-specific kernels constructed between each pair of neurons. This operation fuses weighted information across timesteps, generating a new output spike sequence that aggregates information from two neurons to one.
> * **Regarding the computational cost of larger delay windows:** This does not increase the computational cost. A larger delay window means a larger delay convolution kernel, which results in more left zero-padding, but the number of convolution computations remains unchanged.
> * **Rationale for Choosing the Gaussian Kernel:**
>   - Learnable Temporal Shift: The parameter $μ$ controls the central position of the delay, enabling the model to explicitly learn the delay magnitude. The smoothness of the Gaussian kernel ensures gradient stability during training.
>   - Adjustable Receptive Field: The variance parameter $σ$ modulates the receptive field of the delay kernel, allowing it to capture multi-step historical information.
> * **Network Architecture:** The dynamic graph is divided into fixed-interval snapshots. For each snapshot, second-order neighborhood sampling captures multi-hop dependencies, and first-order aggregation integrates local features. Features are encoded into spike representations using SNNs, converting continuous data into spike trains. Second-order aggregation incorporates broader context. In the Delay-SNN layer, the model dynamically adjusts the Gaussian delay kernel's center to capture varying temporal delays, generating node delay representations. Multi-timescale fusion integrates representations across timesteps, producing final embeddings used for downstream tasks.
> * **SNN-GNN Interaction and Benefits of SNNs for Dynamic Graphs:**
> After traditional GNN-based neighborhood aggregation, the threshold-triggering mechanism of SNNs replaces conventional nonlinear activation functions. Advantages of SNNs for Dynamic Graphs: SNNs inherently accumulate membrane potentials across timesteps, aligning naturally with the discrete timestep nature of dynamic graphs. At each timestep, node information is stored in the membrane potential of SNN neurons. This potential is inherited to the next timestep, preserving historical states and enabling the modeling of graph evolution.
>
> **Response to Weaknesses Theoretical Claims:**
>
> Thank you for your valuable feedback. Your insights have provided us with an important perspective. To further explore this issue, we plan to conduct more detailed experiments focusing on: Evaluating our method's performance on dynamic graphs with diverse temporal patterns and topological structures. Testing our theoretical findings with additional real-world datasets to ensure their validity and applicability in practical scenarios.

---

### Official Review · Reviewer_zQNf · 2025-03-12

**Overall Recommendation:** 2

**Summary:**

This paper proposes Delay-DSGN, a dynamic spiking graph neural network that incorporates a delay convolution kernel, which dynamically adjusts the weight of information at different time steps. Through the delay convolution kernel, Delay-DSGN captures temporal dependencies and historical influences on node representations, then fed into LIF neurons for spike generation. After temporal modeling, Delay-DSGN combines node representations across time steps into a unified representation and finally uses regularization and cross-entropy loss to prevent gradient explosion and vanishing issues.

**Claims And Evidence:**

yes

**Essential References Not Discussed:**

no

**Experimental Designs Or Analyses:**

The experimental design is well-structured, but I have concerns regarding the analysis.

**Methods And Evaluation Criteria:**

yes

**Other Comments Or Suggestions:**

For the formulation, there are three different symbols, please check the Eqs. 4, 9, and 15.

**Other Strengths And Weaknesses:**

**Strengths**

1. This paper considers the historical information in neighborhood aggregation, mitigating historical information forgetting.
2. The delay convolution kernel uses the learnable synaptic delay and weight to make the information of different time steps participate in convolution with different weights.

**Weaknesses**

1. The motivation for delaying the information propagation is not clearly and convincing explained.
2. The fixed random delay model with randomly initialized decay values is unreasonable, since earlier information is more likely to be weighted lower than more recent information.
3. The novelty of this paper is rather limited. Apart from the design of delay convolution kernel, Delay-DSGN is similar to standard SNNs. Moreover, the extensibility of delay convolution kernel is restricted to SNNs.

**Questions For Authors:**

1. When adding a new edge, why doesn't it immediately affect the representations of the two nodes that the edge connects? How can the delay representation ensure that it does not cause information latency, leading to inaccurate node representation?
2. Why the fixed random delay model with randomly initialized decay values outperforms the no-delay model, and sometimes is competitive with Delay-DSGN?
3. Since the delay convolution kernel has already captured the temporal dependencies and historical influences on node representations, why do Delay-DSGN still need to combine node representations from different time steps into a unified feature?
4. Why is the number of neurons set to a higher value in the no-delay model? It will lead to an unfair performance comparison between Delay-DSGN and no-decay model.

**Relation To Broader Scientific Literature:**

A new exploration in spike neural networks for dynamin  graph learning.

**Theoretical Claims:**

yes

---

> ### Author Rebuttal · Authors · 2025-03-31
>
> Thank you for your meticulous review of our work and the valuable feedback provided.
>
> **Response to Weaknesses 1 and Question 1:**
>
>  When a new edge is added to a graph, it may indeed represent an immediate interaction demand. However, in reality, there is usually a time lag between "establishing a connection" and "producing an effect." For instance, in social networks, after user A posts content, user B may take several days to respond; in traffic networks, congestion propagation requires time. These phenomena indicate that delays in information propagation are an inherent property of dynamic graph evolution. Therefore, our motivation is to explicitly model these delays to more accurately capture the dynamics and temporal dependencies. Delay-DSGN adaptively learns the delay parameters $\mu_{ij}$ within the delay kernel, dynamically determining appropriate delay value. This ensures that the speed of information propagation adapts to different application scenarios while preventing excessive delays.
>
> **Response to Weaknesses 2:**
>
> Neither our model nor the fixed random delay model includes a decay value for random initialization. The key difference lies in whether the delay parameters are learnable or remain fixed after initialization.
>
> **Response to Question 2:**
>
> When the number of synaptic connections in a neural network layer far exceeds the number of possible discrete delay positions, randomly initialized delays can cover the entire effective time range. For example, with 10 possible delay positions and 1000 synaptic connections, all delay positions will be covered. Thus, the necessity of moving delay positions away from their initial state diminishes. In this way, a fixed random delay model can achieve good performance by optimizing connection weights to capture temporal characteristics.
>
> **Response to Weaknesses 3:**
>
> * The core innovation of Delay-DSGN lies in its first-ever deep integration of biologically inspired delay mechanisms with dynamic graph modeling, along with rigorous theoretical guarantees for model stability. Specifically, Delay-DSGN explicitly models multi-step dependencies using learnable Gaussian delay kernels, capturing the relationship between topological connection strengths and signal propagation delays in dynamic graphs. Additionally, it provides strict upper bounds for gradient propagation in the spatiotemporal domain driven by delay kernels, offering theoretical guidance for model parameter selection. Compared to traditional SNNs and GNNs, Delay-DSGN not only enhances biological fidelity and temporal processing for SNNs but also introduces a groundbreaking delay perspective to dynamic graph modeling methods.
> * Although current research mainly focuses on SNNs, the delay mechanism can be regarded as a type of temporal convolution method. This approach shows potential for extension to other types of dynamic models when dealing with time-series data. Future work will further explore the application scope of this mechanism.
>
> **Response to Question 3:**
>
> The delay convolution kernel primarily addresses the weighted aggregation of historical information within a single time step. However, topological changes in dynamic graphs often span multiple time steps. By integrating node representations from different time steps into a unified feature, it becomes possible to better capture long-term dependencies and complex topological evolutions. Secondly, this approach compensates for the sparsity of spike signals, mitigating information loss caused by the sparse nature of spike signals.
>
> **Response to Question 4:**
>
> Since Delay-DSGN introduces additional delay time parameters, to balance the total number of parameters, we increased the number of hidden neurons in the no-delay model. Additionally, we conducted experiments on a standard SNN model without adding extra neurons. The comparative experimental results are as follows:
> |  | Metric   | No-Delay SNN | Standard SNN |
> |---------|----------|:--------------:|:--------------:|
> | DBLP    | Ma-F1    | 71.00  | 70.88|
> |         | Mi-F1    | 71.65| 71.98 |
> | Tmall   | Ma-F1    | 59.34| 58.84|
> |         | Mi-F1    | 63.55| 63.52 |
> | Patent  | Ma-F1    | 83.57| 83.53 |
> |         | Mi-F1    | 83.55| 83.48 |
>
> **Response to Other Comments or Suggestions:**
>
> - Equation (4) describes the node features processed with delay, which are subsequently used as input in Equation (5).
> - We understand your questions about Equations (9) and (15) mainly concern the notation $w$，and we apologize for not providing sufficient explanations in the original manuscript. Here are the specific modifications:
>
>     - For Equation (9), $w_t$ represents the weight of the features at time step $t$, which is obtained through an element-wise multiplication of $Z_v^t$ and $w_t$, followed by a row-wise summation.
>
>     - For Equation (15), $w_{ij}$ represents the weight between neuron $i$ and neuron $j$.

---

> > ### Comment · Reviewer_zQNf · 2025-04-01
> >
> > Thank you for your response. I have carefully reviewed your response and intend to increase my rating to 2 based on clarifying the motivation for the delay.
> >
> > AQ-1: You mentioned that "the delay mechanism can be regarded as a type of temporal convolution method." Could you clarify what you mean by "temporal" convolution? How is the term "temporal" defined in this context?
> >
> > AQ-2: Does this paper capture long-term dependencies?

---

> > > ### Author Response · Authors · 2025-04-03
> > >
> > > We are grateful for your feedback and the higher evaluation of our work.
> > >
> > > **Response to Question 1:**
> > >
> > > The data we process is time-series data, where neurons exhibit corresponding spike activity (0 or 1) at each time step. Arranging these spike activities in the order of time steps constitutes the spike sequences mentioned in the text.
> > > $$
> > > \widetilde{s_v^{j,\ t}}=\ \left[0,\ 0,\ \ldots,\ 0,\ s_v^{j,t}\right]
> > > $$
> > > The delayed convolutional kernel is defined as:
> > > $$
> > > k_{ij}\left[n\right]=w_{ij}exp\left(\frac{-\left(n-\left(K_s-d_{ij}-1\right)\right)^2}{2\left(\sigma\right)^2}\right)
> > > $$
> > > Here, $d_{ij}$ represents the delay amount, and $K_s-d_{ij}-1$ corresponds to the index in the convolutional kernel that assigns high weight.
> > > Temporal convolution refers to the process where the convolutional kernel slides over the spike sequence (along the time axis) to extract features, resulting in the input to the SNN:
> > > $$
> > > I_v^{j,\ t}=k_{ij}\ast\widetilde{s_v^{j,\ t}}
> > > $$
> > > This process is analogous to 2D convolution in image processing, except that in our work, it is performed on one-dimensional time series data.
> > >
> > > **Response to Question 2:**
> > >
> > > Delay-DSGN successfully captures long-term dependencies. On the DBLP, we observed the weights of 10 features across 27 time steps. The results indicate that the model assigns higher weights at earlier time steps, as shown in the table below. Moreover, compared to SpikeNet and Dy-SIGN, which only use information from the previous time step to update the membrane potential, our delayed kernel aggregates multi-step historical information to update the membrane potential of the current neuron. The results of comparative experiments also demonstrate that our method outperforms these two traditional SNN+GNN approaches in terms of performance.
> > > | Time Step | feature1-weight | feature2-weight | feature3-weight | feature4-weight | feature5-weight | feature6-weight | feature7-weight | feature8-weight | feature9-weight | feature10-weight |
> > > |---------------------|-----------|-----------|-----------|-----------|-----------|-----------|-----------|-----------|-----------|------------|
> > > | 1                   | 0.139       | 0.205       | 0.247       | -0.058       | 0.102       |0.034       | 0.252       | 0.146       | 0.154       | 0.090        |
> > > | 5                   | 0.114      | 0.153      | 0.093      | 0.131      | -0.011      | -0.163      | -0.072      | -0.013      | 0.171      | 0.031       |
> > > | 10                  | 0.054      | 0.026      | 0.012      | 0.058      | 0.022      | 0.050      | 0.027      | 0.062      | 0.021      | 0.068       |
> > > | 15                  | -0.008      | -0.135      | 0.028      | -0.045      | 0.012      | 0.130      | -0.017      | 0.007      | 0.034     | -0.218       |
> > > | 20                  | 0.077      | -0.087      | -0.134      | 0.009      | 0.056      | 0.076      | -0.076      | -0.076      | 0.029      |-0.103       |
> > > | 25                  | 0.143       | 0.123      | 0.276      | 0.031       | -0.073       | 0.106      | -0.180      | -0.158       | -0.001      | 0.094        |
> > > | 27                  | 0.349      | 0.421      | 0.370      | -0.179      | 0.155      | 0.244      | -0.070      | -0.126      | -0.184      | 0.175       |
> > >
> > > Thank you again for your recognition of our work.

---

### Decision · Program_Chairs · 2025-05-01

**Decision:**

Accept (poster)

**Comment:**

This paper proposes a dynamic spiking graph neural network to capture the impact of latency in information propagation with a learnable mechanism. It dynamically adjusts connection weights and propagation speeds with a Gaussian delay kernel to adaptively delay historical information. Theoretical and experimental justifications demonstrate its superiority in high-performance and stable training processes. The claims are supported by clear and convincing evidence.  This paper possesses a novel delay-based temporal mechanism and solid theoretical justification.  In the rebuttal, the authors try to clarify the motivation for the delay.